# Evaluating poverty alleviation strategies in a developing country

**Pramod K. Singh** [ID]*, **Harpalsinh Chudasama**

Institute of Rural Management Anand (IRMA), Anand, Gujarat, India

* pramod@irma.ac.in

## Abstract

A slew of participatory and community-demand-driven approaches have emerged in order to address the multi-dimensional nature of poverty in developing nations. The present study identifies critical factors responsible for poverty alleviation in India with the aid of fuzzy cognitive maps (FCMs) deployed for showcasing causal reasoning. It is through FCM-based simulations that the study evaluates the efficacy of existing poverty alleviation approaches, including community organisation based micro-financing, capability and social security, market-based and good governance. Our findings confirm, to some degree, the complementarity of various approaches to poverty alleviation that need to be implemented simultaneously for a comprehensive poverty alleviation drive. FCM-based simulations underscore the need for applying an integrated and multi-dimensional approach incorporating elements of various approaches for eradicating poverty, which happens to be a multi-dimensional phenomenon. Besides, the study offers policy implications for the design, management, and implementation of poverty eradication programmes. On the methodological front, the study enriches FCM literature in the areas of knowledge capture, sample adequacy, and robustness of the dynamic system model.

**Data Availability Statement:** All relevant data are within the manuscript and its Supporting Information files. The aggregated condensed matrix (social cognitive map) is given in S1 Table. One can replicate the findings of this study by analyzing this weight matrix.

## 1. Introduction

### 1.1. Poverty alleviation strategies

Although poverty is a multi-dimensional phenomenon, poverty levels are often measured using economic dimensions based on income and consumption [1]. Amartya Sen's capability deprivation approach for poverty measurement, on the other hand, defines poverty as not merely a matter of actual income but an inability to acquire certain minimum capabilities [2]. Contemplating this dissimilarity between individuals' incomes and their inabilities is significant since the conversion of actual incomes into actual capabilities differs with social settings and individual beliefs [2–4]. The United Nations Development Programme (UNDP) also emphasises the capabilities' approach for poverty measurement as propounded by Amartya Sen [5]. "*Ending poverty in all its forms everywhere*" is the first of the 17 sustainable development goals set by the United Nations with a pledge that no one will be left behind [6]. Development projects and poverty alleviation programmes all over the world are predominantly aimed

**Funding:** The World Bank and the Ministry of Rural Development, Government of India

**Competing interests:** We hereby declare that none of the authors have any conflict of interest on this paper.

at reducing poverty of the poor and vulnerable communities through various participatory and community-demand-driven approaches [7,8]. Economic growth is one of the principal instruments for poverty alleviation and for pulling the poor out of poverty through productive employment [9,10]. Studies from Africa, Brazil, China, Costa Rica, and Indonesia show that rapid economic growth lifted a significant number of poor people out of financial poverty between 1970 and 2000 [11]. According to Bhagwati and Panagariya, economic growth generates revenues required for expanding poverty alleviation programmes while enabling governments to spend on the basic necessities of the poor including healthcare, education, and housing [9]. Poverty alleviation strategies may be categorised into four types including community organisations based micro-financing, capability and social security, market-based, and good governance.

Micro-finance, aimed at lifting the poor out of poverty, is a predominant poverty alleviation strategy. Having spread rapidly and widely over the last few decades, it is currently operational across several developing countries in Africa, Asia, and Latin America [12–21]. Many researchers and policy-makers believe that access to micro-finance in developing countries empowers the poor (especially women) while supporting income-generating activities, encouraging the entrepreneurial spirit, and reducing vulnerability [15, 21–25]. There are fewer studies, however, that show conclusive and definite evidence regarding improvements in health, nutrition, and education attributable to micro-finance [21,22]. For micro-finance to be more effective, services like skill development training, technological support, and strategies related to better education, health and sanitation, including livelihood enhancement measures need to be included [13,17,19].

Economic growth and micro-finance for the poor might throw some light on the financial aspects of poverty, yet they do not reflect its cultural, social, and psychological dimensions [11,21,26]. Although economic growth is vital for enhancing the living conditions of the poor, it does not necessarily help the poor exclusively tilting in favour of the non-poor and privileged sections of society [4]. Amartya Sen cites social exclusion and capability deprivation as reasons for poverty [4,27]. His capabilities' approach is intended to enhance people's well-being and freedom of choices [4,27]. According to Sen, development should focus on maximising the individual's ability to ensure more freedom of choices [27,28]. The capabilities approach provides a framework for the evaluation and assessment of several aspects of the individual's well-being and social arrangements. It highlights the difference between means and ends as well as between substantive freedoms and outcomes. An example being the difference between fasting and starving [27–29]. Improving capabilities of the poor is critical for improving their living conditions [4,10]. Improving individuals' capabilities also helps in the pooling of resources while allowing the poor to engage in activities that benefit them economically [4,30]. Social inclusion of vulnerable communities through the removal of social barriers is as significant as financial inclusion in poverty reduction strategies [31,32]. Social security is a set of public actions designed to reduce levels of vulnerability, risk, and deprivation [11]. It is an important instrument for addressing the issues of inequality and vulnerability [32]. It also induces gender parity owing to the equal sovereignty enjoyed by both men and women in the context of economic, social, and political activities [33].

The World Development Report 1990 endorsed a poverty alleviation strategy that combines enhanced economic growth with provisions of essential social services directed towards the poor while creating financial and social safety nets [34,35]. Numerous social safety net programmes and public spending on social protection, including social insurance schemes and social assistance payments, continue to act as tools of poverty alleviation in many of the developing countries across the world [35–39]. These social safety nets and protection programmes show positive impacts on the reduction of poverty, extent, vulnerability, and on a wide range

of social inequalities in developing countries. One major concern dogging these programmes, however, is their long-term sustainability [35].

Agriculture and allied farm activities have been the focus of poverty alleviation strategies in rural areas. Lately, though, much of the focus has shifted to livelihood diversification on the part of researchers and policy-makers [15,40]. Promoting non-farm livelihoods, along with farm activities, can offer pathways for economic growth and poverty alleviation in developing countries the world over [40–44]. During the early 2000s, the development of comprehensive value chains and market systems emerged as viable alternatives for poverty alleviation in developing countries [45]. Multi-sectoral micro-enterprises may be deployed for enhancing productivity and profitability through value chains and market systems, they being important for income generation of the rural poor while playing a vital role in inclusive poverty eradication in developing countries [46–48].

Good governance relevant to poverty alleviation has gained top priority in development agendas over the past few decades [49,50]. Being potentially weak in the political and administrative areas of governance, developing countries have to deal with enormous challenges related to social services and security [49,51]. In order to receive financial aid from multinational donor agencies, a good governance approach towards poverty reduction has become a prerequisite for developing countries [49,50]. This calls for strengthening a participatory, transparent, and accountable form of governance if poverty has to be reduced while improving the lives of the poor and vulnerable [50,51]. Despite the importance of this subject, very few studies have explored the direct relationship between good governance and poverty alleviation [50,52,53]. Besides, evidence is available, both in India and other developing countries, of information and communication technology (ICT) contributing to poverty alleviation programmes [54]. Capturing, storing, processing, and transmitting various types of information with the help of ICT empowers the rural poor by increasing access to micro-finance, expanding the use of basic and advance government services, enabling the development of additional livelihood assets, and facilitating pro-poor market development [54–56].

## 1.2. Proposed contribution of the paper

Several poverty alleviation programmes around the world affirm that socio-political inclusion of the poor and vulnerable, improvement of social security, and livelihood enhancement coupled with activities including promoting opportunities for socio-economic growth, facilitating gender empowerment, improving facilities for better healthcare and education, and stepping up vulnerability reduction are central to reducing the overall poverty of poor and vulnerable communities [1,11]. These poverty alleviation programmes remain instruments of choice for policy-makers and development agencies even as they showcase mixed achievements in different countries and localities attributable to various economic and socio-cultural characteristics, among other things. Several poverty alleviation programmes continue to perform poorly despite significant investments [8]. The failure rate of the World Bank's development projects was above 50% in Africa until 2000 [57]. Hence, identifying context-specific factors critical to the success of poverty alleviation programmes is vital.

Rich literature is available pertinent to the conceptual aspects of poverty alleviation. Extant literature emphasises the importance of enhancing capabilities and providing social safety, arranging high-quality community organisation based micro-financing, working on economic development, and ensuring good governance. However, the literature is scanty with regard to comparative performances of the above approaches. The paper tries to fill this gap. This study, through fuzzy cognitive mapping (FCM)-based simulations, evaluates the efficacy of these approaches while calling for an integrative approach involving actions on all dimensions to

eradicate the multi-dimensional nature of poverty. Besides, the paper aims to make a two-fold contribution to the FCM literature: i) knowledge capture and sample adequacy, and ii) robustness of the dynamic system model.

The remainder of the paper proceeds as follows: We describe the methodology adopted in the study in section two. Section three illustrates key features of the FCM system in the context of poverty alleviation, FCM-based causal linkages, and policy scenarios for poverty alleviation with the aid of FCM-based simulations. We present our contribution to the extant literature relating to FCM and poverty alleviation. Finally, we conclude the paper and offer policy implications of the study.

## 2. Methodology

We conducted the study with the aid of the FCM-based approach introduced by Kosko in 1986 [58]. The process of data capture in the FCM approach is considered quasi-quantitative because the quantification of concepts and links may be interpreted in relative terms [59] allowing participants to debate the cause-effect relations between the qualitative concepts while generating quantitative data based on their experiences, knowledge, and perceptions of inter-relationships between concepts [60–64,65–68]. The FCM approach helps us visualise how interconnected factors/ variables/ concepts affect one another while representing self-loop and feedback within complex systems [62,63,69]. A cognitive map is a signed digraph with a series of feedback comprising concepts (nodes) that describe system behavior and links (edges) representing causal relationships between concepts [60–63,65,70–72]. FCMs may be created by individuals as well as by groups [60,72,73]. Individual cognitive mapping and group meeting approaches have their advantages and drawbacks [72]. FCMs allow the analysis of non-linear systems with causal relations, while their recurrent neural network behaviour [69,70,74] help in modelling complex and hard-to-model systems [61–63]. The FCM approach also provides the means to build multiple scenarios through system-based modelling [60–64,69,74,75].

The strengths and applications of FCM methodology, focussing on mental models, vary in terms of approach. It is important to remember, though, that (i) the FCM approach is not driven by data unavailability but is responsible for generating data [60,76]. Also that (ii) FCMs can model complex and ambiguous systems revealing hidden and important feedback within the systems [58,60,62,69,76] and (iii) FCMs have the ability to represent, integrate, and compare data–an example being expert opinion vis-à-vis indigenous knowledge–from multiple sources while divulging divergent viewpoints [60]. (iv) Finally, FCMs enable various policy simulations through an interactive scenario analysis [60,62,69,76].

The FCM methodology does have its share of weaknesses. To begin with: (i) Respondents' misconceptions and biases tend to get encoded in the maps [60,62]. (ii) Possibility of susceptibility to group power dynamics in a group model-building setting cannot be ruled out; (iii) FCMs require a large amount of post-processing time [67]. (iv) The FCM-based simulations are non-real value and relative parameter estimates and lack spatial and temporal representation [60,77,78].

These drawbacks notwithstanding, we, along with many researchers, conceded that the strengths and applications of FCM methodology outweighed the former, particularly with regard to integrating data from multiple stakeholders with different viewpoints.

We adopted the multi-step FCM methodology discussed in the following sub-sections. We adopted the multi-step FCM methodology discussed in the following sub-sections. We obtained individual cognitive maps from the participants in two stages: 'open-concept design' approach followed by the 'pre-concept design' approach. We coded individual cognitive maps

into adjacency matrices and aggregated individual cognitive maps to form a social cognitive map. FCM-based simulation was used to build policy scenarios for poverty alleviation using different input vectors.

## 2.1. Obtaining cognitive maps from the participants

A major proportion of the literature on fuzzy cognitive maps reflects an 'open-concept design' approach, while some studies also rely on a 'pre-designed concept' approach with regard to data collection.

In the case of the 'open-concept design' approach, concepts are determined entirely by participants and are unrestricted [59,60,62,63,65–67,79,80]. While the researcher determines the context of the model by specifying the system being modelled, including the boundaries of the system, participants are allowed to decide what concepts will be included. This approach provides very little restriction in the knowledge capture from participants and can be extremely beneficial especially if there is insufficient knowledge regarding the system being modelled.

In the case of the 'pre-designed concept' approach, concepts are pre-determined either by experts or by researchers using available literature [64,69,74,81,82]. In this approach, the researcher is able to exercise a higher degree of control over how the system is defined. The 'pre-designed concept' approach is likely to be more efficient compared to the 'open-concept design' in the context of time required for model building. However, it restricts the diversity of knowledge captured from participants and is able to influence more heavily the way in which this knowledge is contextualised based on input and interpretation.

We have adopted a 'mixed-concept design' approach for this study involving data collection in two stages:

**2.1.1. Stage one: 'Open-concept design' approach.** During the first stage, we engaged with the experts and national-level policy-makers who designed the *Deendayal Antyodaya Yojana*-National Rural Livelihoods Mission (DAY-NRLM), a centrally sponsored programme in India. The DAY-NRLM aims at abolishing rural poverty by promoting multiple livelihoods for the rural poor and vulnerable households. The programme is focussed on organising the rural poor and vulnerable communities into self-help groups (SHGs) while equipping them with means of self-employment. The four critical components of the programme *viz.*, (i) universal social mobilisation and institution building, (ii) financial inclusion, (iii) convergence and social development, and (iv) livelihood enhancement are designed to address the exclusions of these communities, eliminate their poverty, and bring them within the ambit of mainstream economic and social systems. Participants comprising three experts from the World Bank, nine experts from the National Mission Management Unit of the DAY-NRLM, and 25 monitoring and evaluation experts from 25 states of India created 37 FCMs. A sample map of FCMs obtained from these participants is provided in S1 Fig. We demonstrated the construction of fuzzy cognitive maps with the aid of a map from a neutral problem domain referring to direct and consequential impacts of deforestation, which had been approved by the 'Research Ethics Committee' of our Institute.

A group discussion was held with the participants regarding the issues under investigation subsumed under the title "critical factors required to ensure that people come out of poverty on a sustainable basis". It prompted them to identify major concepts pertaining to the above. These were listed down on a whiteboard by the researchers. Once the participants had understood the process of drawing a fuzzy cognitive map and identified major concepts responsible for poverty alleviation, they were asked to draw a fuzzy cognitive map individually. The participants used the concepts listed on the whiteboard to draw fuzzy cognitive maps. Many participants added new concepts while drawing the maps. They then connected all the concepts

through various links based on their personal understanding. The links, represented by arrows in between concepts, show the direction of influence between them.

The participants assigned weights to each link on a scale of 1–10 to describe the relationship strength between two concepts [60]. Ten denoted the highest strength and one the lowest; the numbers 1–3 signified relationships with low strength, 4–6 signified relationships with medium strength, and 7–10 signified relationships with high strength. After constructing the FCMs each participant made a presentation, which was video-recorded, explaining their map to the researchers. The researchers, based on causal relationships between the concepts, assigned positive and negative polarities to the weights of the links [59,60,62–64,66–68,72].

**2.1.2. Stage two: 'Pre-designed concept' approach.** During the second stage, an instrument depicting 95 concepts under 22 concept categories was prepared based on the FCMs obtained from participants during the first stage (S2 Fig). The instrument also contained links between the 22 concept categories. 'Research Ethics Committee' of our Institute approved this instrument as well. We used this instrument during the second stage to obtain FCMs. We obtained 123 additional FCMs, of which 20 FCMs were obtained from the Chief Executive Officers along with experts from livelihood, enterprise, and community development domains belonging to the National Mission Management Unit in the states of Bihar, Jharkhand, Madhya Pradesh, and Maharashtra. The remaining 103 FCMs were obtained from 103 district project coordinators, who had agreed to participate in the study. Unlike most FCM-based studies, which usually rely upon 30 to 50 participants, this study involved 174 experts and project implementers. Most participants produced FCMs individually and some in pairs. The 174 participants produced 160 FCMs.

The participants were given the instrument and were instructed to assign weights to each concept, wherever applicable, and leave other cells blank. These weights were assigned based on the concepts' significance regarding poverty alleviation in India. The instrument was designed to allow participants to add new concepts and/or remove existing ones from the instrument based on their understanding and perceptions. Later, the participants were asked to assign weights to all pre-established links between the 22 concept categories. The instrument also allowed participants to draw new linkages between the categories and/or discard the existing relationships based on their understanding and perceptions. After constructing the FCMs each participant made a presentation to the researchers, which was video-recorded. During the process, participants added 55 new concepts within the pre-classified 22 concept categories. Five new concepts were added under a new category. The final data comprised 23 concept categories and 155 concepts (S3 Fig).

## 2.2. Coding individual cognitive maps into adjacency matrices

The individual FCMs were coded into separate excel sheets, with concepts listed in vertical and horizontal axes, forming an N x N adjacency matrix. The weights of the links, on a scale of 1–10, were normalised in the −1 to +1 range [62,63]. The values were then coded into a square adjacency matrix whenever a connection existed between any two concepts [60,62–64,66].

## 2.3. Aggregation of individual cognitive maps

There are various methods of aggregating individual FCMs; each method has advantages and disadvantages [83]. We aggregated individual adjacency matrices obtained by normalising each adjacency matrix element according to its decisional weight, $w_i$, and the number of participants, $k$, who supported it. The following equation illustrates the augmentation of

individual adjacency matrices:

$$M_{FCM} = \sum_{i=1}^{k} \frac{w_i m_i}{k} \tag{1}$$

$M_{FCM}$ is the aggregated adjacency matrix, where, $k$ represents the number of participants interviewed; $w_i$ is the decisional weight of the expert $i$, where, $\sum_{i=1}^{k} w_i = 1$; and $m_i$ is the adjacency matrix written by the participant $i$.

This aggregation approach has been adopted by many researchers [59,60,63–67,74,79,84–87]. A large number of concepts in an aggregated (social/ group) fuzzy cognitive map with many interconnections and feedback form a complex system. Aggregation of all the 160 individual cognitive maps produced a social cognitive map (S1 Table). This shows the cumulative strength of the system.

## 2.4. Structural analysis of the system

Structural analysis of the final condensed social cognitive map was undertaken using the FCMapper software. The graph theory of a cognitive map provides a way of characterising FCM structures employing several indices in addition to the number of concepts (C) and links (W) such as in-degree, out-degree, centrality, complexity index, and density index [60].

The in-degree is the column sum of absolute values of a concept in the adjacency matrix. It shows the cumulative strength of links entering the concept ($w_{ji}$). Where $n$ = the total number of concepts:

$$\text{In−degree} = \sum_{j=1}^{n} w_{ji} \tag{2}$$

The out-degree is the row sum of absolute values of a concept in the adjacency matrix. It shows the cumulative strengths of links exiting the concept ($w_{ij}$). Where $n$ = the total number of concepts:

$$\text{Out−degree} = \sum_{j=1}^{n} w_{ij} \tag{3}$$

The degree centrality of a concept is the summation of its in-degree and out-degree. The higher the value, the greater is the importance of a concept in the overall model [60].

Transmitter concepts (T) depict positive out-degree and zero in-degree. Receiver concepts (R) represent positive in-degree and zero out-degree. Ordinary concepts (O) have both a non-zero in-degree and out-degree [60].

The complexity index of a cognitive map is the ratio of receiver concepts (R) to transmitter concepts (T). Higher complexity indicates more complex systems thinking [60]:

$$Complexity = {R}/{T} \tag{4}$$

The density index of a cognitive map is an index of connectivity showing how connected or sparse the maps are. It is a product of the number of concepts (C) and the number of links (W). Here the number of existing links is compared to the number of all possible links. Higher the density, greater the existence of potential management policies [60]:

$$Density = \frac{W}{C(C-1)} \tag{5}$$

## 2.5. Fuzzy cognitive maps-based simulations

The scenarios formed through FCM-based simulations can serve to guide managers and policy-makers during the decision-making process [62–64,66,69,82,88–90]. An FCM is formed out of the adjacency matrix and a state vector, representing the values of the connections between the concepts and the values of the system concepts [62,63,69]. The weighted adjacency matrix of an FCM forms a recurrent neural network, including concepts and interconnections for processing the information and feedback loops [88,91]. These have been used to analyse system behavior by running FCM-based simulations in order to determine possible future scenarios.

In order to understand FCM-based simulations, let us understand the FCM as a quadruple, i.e. $M = (C_n, W, A, f)$, where, $n$ is the set of all concepts ($C$) in the map, $W$: $(C_i, C_j) \rightarrow w_{ij}$ is a function which defines the causal weight matrix, $W_{M \times M}$, $A$: $(C_i) \rightarrow A^{(t)}_i$ is a function that computes the activation degree of each concept $C_i$ at the discrete-time step $t$ ($t = 1, 2, \ldots, T$), and $f$ (.) is the transfer function [63,71,92,93]. Knowledge and experience of stakeholders regarding the system determine the type and number of concepts as well as the weights of the links in FCMs. The value $A_i$ of a concept $C_i$, expresses the quantity of its corresponding value. With values assigned to the concepts and weights, the FCM converges to an equilibrium point [71,91]. At each step, the value $A_i$ of a concept is calculated, following an activation rule, which computes the influence of other concepts to a specific concept.

We have used an increasingly popular activation rule [61–64,90,91] introduced by Stylios [94], which is as follows:

$$A_i^{(t+1)} = f\left(A_i^{(t)} + \sum_{j=1}^{n} w_{ji} \times A_j^{(t)}\right), i \neq j \tag{6}$$

Where, $n$ is the total number of concepts, $A_i^{(t+1)}$ is the value of concept $C_i$ at simulation step $t + 1$, $A_i^{(t)}$ is the value of concept $C_i$ at simulation step $t$, $A_j^{(t)}$ is the value of concept $C_j$ at simulation step $t$, $w_{ji}$ is the weight of the interconnection between concept $C_j$ and concept $C_i$, and $f$ is the transformation function [64,90]. The restriction $i \neq j$ is used when self-causation is assumed to be impossible [91].

The simulation outcomes also depend on the type of transformation function used. The most frequently used transformation functions ($f$) are sigmoid and hyperbolic tangent functions [90–93]. When the values of concepts can only be positive, i.e. in the range of (0,1), the most common unipolar sigmoid transformation function is used [64,91–93]. Following is the mathematical equation of the sigmoid transformation function:

$$f(x) = \frac{1}{1 + e^{-\lambda x}} \tag{7}$$

Where, $\lambda$ is a real positive number ($\lambda > 0$) and a constant value that determines the slope steepness factor, while $x$ is the value of concept $A_i^{(t)}$ on the equilibrium point [64,93]. Higher values of $\lambda$ increase the steepness and make it more sensitive to the changes of $x$. Hence, the derivative $\delta f / \delta x$ becomes higher when increasing the activation value [95].

**2.5.1. Development of input vectors for policy scenarios.** Identifying pivotal concepts is a traditional approach in scenario planning that helps linking storylines to the quantitative model [96]. In the FCM-based scenario analysis, recognition of such pivotal concepts, termed as input vectors, mainly relies upon participants' perceptions along with the characteristics of the model. We identified four input vectors for four poverty alleviation policy scenarios based on existing literature on poverty alleviation strategies. The fifth input vector is based on the concepts with the highest weights identified by the participants. In the sixth input vector, the

concept representing entrepreneurship is replaced by the concept representing livelihood diversification considering its importance based on existing literature [15,40]. All six scenarios are explained below:

*Scenario 1*: *High-quality community organisation based micro-financing*—Input vector 1: C2, C3, C4, C5, C11, and C12 (strong institutions of the poor, community heroes driving the programme, capacity building of the community organisations, mainstream financial institutions supporting community organisations, need-based finance, and developing repayment culture). This scenario tries to examine how high-quality community organisation based micro-finance could alleviate poverty.

*Scenario 2*: *Capabilities and social security*—Input vector 2: C19, C20, C21, C22, and C23 (affordable and approachable education and healthcare, social inclusion, building personal assets, adequate knowledge base, and vulnerability reduction). This scenario tries to estimate how improving the capabilities of the poor and providing them social security would help alleviate poverty.

*Scenario 3*: *Market-based approach*—Input vector 3: C13, C14, C15, C16, and C17 (livelihood diversification, entrepreneurship, multi-sectoral collective enterprise, value addition by collectives, and market linkages). This scenario tries to evaluate how a market-based approach could alleviate poverty.

*Scenario 4*: *Good governance approach*—Input vector 4: C6, C7, C8, C9, and C10 (good governance systems and processes, robust monitoring mechanisms, implementation process, linkages/ convergence/ partnerships, and enabling policy & political will). This scenario tries to evaluate how good governance is crucial for poverty alleviation.

*Scenario 5*: *Integrative approach 1*—Input vector 5: C2, C3, C6, C9, C10, C14, and C19 (strong institutions of the poor, community heroes driving the programme, sound governance systems and processes, enabling policy & political will, linkages/ convergence/ partnerships, entrepreneurship, and affordable and approachable education and healthcare). This scenario tries to assess how the most critical concepts, identified by the participants, are crucial for poverty alleviation.

*Scenario 6*: *Integrative approach 2*—Input vector 6: C2, C3, C6, C9, C10, C13, and C19 (strong institutions of the poor, community heroes driving the programme, good governance systems and processes, enabling policy & political will, linkages/ convergence/ partnerships, livelihood diversification, and affordable and approachable education and healthcare). This scenario tries to assess how the most important concepts, including livelihood diversification, are critical for the alleviation of poverty. Based on the relative weights, scenarios 4 to 6 also had alternative input vectors incorporating sensitive support structure (C1) without any demonstrable results.

**2.5.2. Simulation process.**   Each concept in the system has an initial state vector $A_0$ that varies from 0 to |1|. which is associated with an activation vector, where 0 means 'non-activated' and |1| means 'activated' [65,80]. A new state of the concepts can be calculated by multiplying the adjacency matrix with the state vector [69]. When one or more concepts are 'activated' this activation spreads through the matrix following the weighted relationships. During the simulation process, each iteration produces a new state vector with 'activated' concepts and 'non-activated' concepts. Self-loops and feedback cause a repeated activation of concepts, introducing non-linearity to the model [61,70,88]. The activation of concepts is iterated, using a 'squashing function' to rescale concept values towards |1|, until the vector values stabilise and the model reaches equilibrium or steady-state [61,65,70]. The resulting concept values may be used to interpret outcomes of a particular scenario and to study the dynamics of the modeled system [61–63,70].

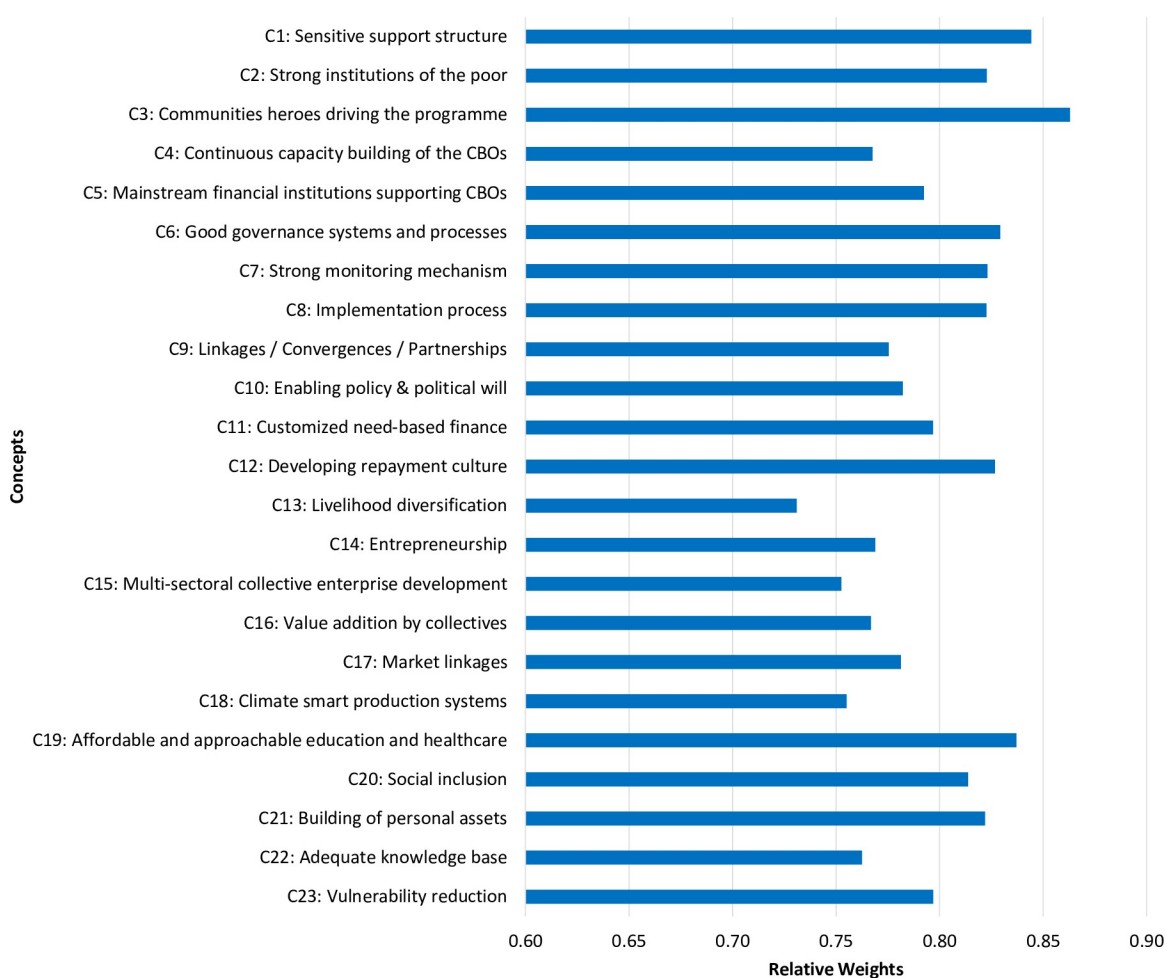

**Fig 1. The relative importance of the concepts as perceived by the stakeholders.**

The simulation process is carried out with the initial state vector of the input vectors, identified in each scenario (1 to 6), clamped to 1 ($A_1$) and the initial state vector of all the other concepts clamped to 0 ($A_0$). We applied the activation rule proposed by Stylios [94], to run simulations because of its memory capabilities along with the sigmoid transformation function as the links have only positive values. The sensitivity of the system was analysed by clamping the concepts of each input vector to 0.1, 0.2, 0.3, 0.4, 0.5, 0.6, 0.7, 0.8, and 0.9 (S4 Fig) to determine whether the system behaves in a similar manner in each simulation [62,63,72,89].

## 3. Results and discussions

### 3.1. Key features of the FCM system in the context of poverty alleviation

The social cognitive map built by combining the individual FCMs comprises 23 concepts and 51 links (Fig 2 and S1 Table). This FCM system has a density index of 0.088, which signifies that 8.8% links are actually made of the maximum number of links that could theoretically exist between the 24 concepts. The FCM system has a complexity index of 0.125, which showcases more utility outcomes and less controlling forcing functions. However, unless the density and complexity values of the FCM system are compared to those of other FCM systems representing a similar topic, interpretation of these figures is challenging [75].

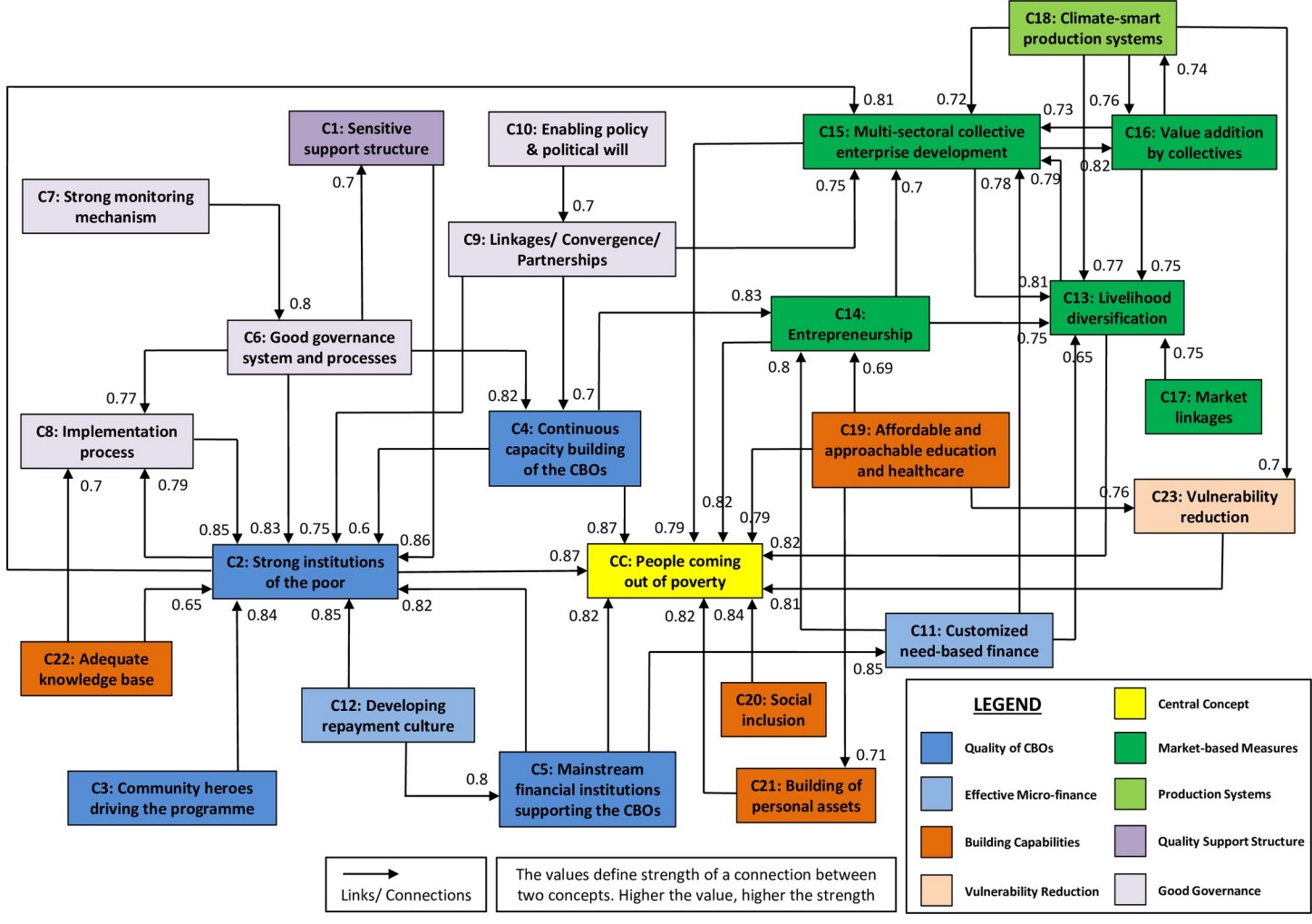

**Fig 2. A social cognitive map showing the critical factors required for poverty alleviation.**

There are some autonomous concepts virtually disengaged from the system. Some dependent concepts although have a relatively low degree of influence, exhibit strong dependence. The contribution of a concept in a cognitive map can be understood by its degree centrality, which is the summation of in-degree and out-degree. Table 1 illustrates the in-degree and out-degree and degree centrality of the FCM system. Concepts have been depicted such as C2: strong institutions of the poor, C15: multi-sectoral collective enterprise development, C13: livelihood diversification and C14: entrepreneurship have higher degree centrality. These concepts should be interpreted as the greatest strength of poverty alleviation strategies. The most influential concepts (i.e., those with the highest out-degree) affecting the poverty alleviation strategies are C6: good governance systems and processes, C19: affordable and approachable education and healthcare, C18: climate-smart production systems, C2: strong institutions of the poor, and C5: mainstream financial institutions supporting CBOs. Scenario analysis results will later help us gain a deeper understanding of the connectivity and influencing concepts of poverty alleviation.

**Table 1. The structural analysis of the FCM system.**

| Concepts | In-degree | Out-degree | Degree Centrality |
|---|---|---|---|
| CC: People coming out of poverty | 8.25 | 0 | 8.25 |
| C1: Sensitive support structure | 0.70 | 0.86 | 1.56 |
| C2: Strong institutions of the poor | 7.04 | 2.47 | 9.51 |
| C3: Community heroes driving the programme | 0 | 0.84 | 0.84 |
| C4: Continuous capacity building of the CBOs | 1.52 | 2.30 | 3.82 |
| C5: Mainstream financial institutions supporting CBOs | 0.80 | 2.49 | 3.29 |
| C6: Good governance systems and processes | 0.80 | 3.11 | 3.91 |
| C7: Strong monitoring mechanism | 0 | 0.80 | 0.80 |
| C8: Implementation process | 2.26 | 0.85 | 3.11 |
| C9: Linkages / Convergences / Partnerships | 0.70 | 2.20 | 2.90 |
| C10: Enabling policy and political will | 0 | 0.70 | 0.70 |
| C11: Customized need-based finance | 0.85 | 2.22 | 3.07 |
| C12: Developing repayment culture | 0 | 1.65 | 1.65 |
| C13: Livelihood diversification | 4.48 | 1.61 | 6.09 |
| C14: Entrepreneurship | 2.31 | 2.27 | 4.58 |
| C15: Multi-sectoral collective enterprise development | 5.28 | 2.42 | 7.70 |
| C16: Value addition by collectives | 1.58 | 2.23 | 3.81 |
| C17: Market linkages | 0 | 0.75 | 0.75 |
| C18: Climate smart production systems | 0.74 | 2.95 | 3.69 |
| C19: Affordable and approachable education and healthcare | 0 | 2.97 | 2.97 |
| C20: Social inclusion | 0 | 0.84 | 0.84 |
| C21: Building of personal assets | 0.73 | 0.82 | 1.55 |
| C22: Adequate knowledge base | 0 | 1.35 | 1.35 |
| C23: Vulnerability reduction | 1.46 | 0.81 | 2.27 |

The participants also provided the state vector values (A) of all the concepts (C) based on their understanding of the relative significance of these concepts regarding poverty alleviation in India (Fig 1).

The results show that participants assigned greater significance to the following concepts- C3: community heroes driving the programme, C1: quality support structure, C19: affordable and approachable education and healthcare, C6: good governance systems and processes, C2: strong institutions of the poor, C12: developing repayment culture, and C7: robust monitoring mechanisms.

The results acknowledge that building strong institutions of the poor for a community-demand-driven and community-managed poverty alleviation programme is likely to enjoy greater success. They also confirm that developing robust monitoring mechanisms can ensure better functioning of the community-based organisations (CBOs). Robust governance systems and processes are essential for vibrant CBOs. They can empower communities to have better access to affordable education and healthcare facilities. Better access to micro-finance for these CBOs could help alleviate the economic poverty of the poor and vulnerable communities.

The results, however, fail to capture the cultural and social dimensions of poverty.

## 3.2. Understanding the poverty alleviation strategy

This section summarises the views of participants across the concepts based on the presentations made by them to the researcher during both the stages of knowledge capture. Fig 2 illustrates the cognitive interpretive diagram formed using the social cognitive map. The concepts,

represented by each node in the diagram, are connected by several links. These links establish relationships between the concepts representing the basis of degree centrality. The central concept is people coming out of poverty, which is depicted with yellow color in Fig 2.

Participants indicated that setting up a quality and dedicated support structures at multiple levels (national, state, district, and block) is essential for poverty alleviation (Fig 2: C1). The support structures should be staffed with professionally competent and dedicated human resources. The crucial role of these support structures is to build and nurture strong institutions of the poor (Fig 2: C2) at multiple levels and evanesce when community heroes start driving the programme. Building and sustaining strong, inclusive, self-managed, and self-reliant institutions of the poor at various levels such as self-help groups (SHGs), village organisations (VOs), and cluster-level federations (CLFs) through training, handholding, and systematic guidance are crucial to the success of a poverty alleviation programme. However, superior CBOs are required to ensure the quality of primary-level institutions and their sustainability. Adherence to the five principles (regular meetings, regular savings, regular inter-loaning, timely repayment of the loans, and up-to-date books of accounts), co-ordination, and cohesiveness between the members would go a long way in building strong institutions of the poor.

Participants emphasised the importance of community heroes in driving the poverty alleviation programme (Fig 2: C3). The poverty eradication programme is likely to meet with greater success if it is entirely operated and managed by the community. Involving experienced community members for social mobilisation, capacity building and scaling-up of various processes within the project will ensure effective functioning and implementation of the programme. Participants believed that the capacity building of the CBOs, community resource persons, community cadres, and community service providers (Fig 2: C4) are essential for poverty alleviation. Apart from training in social and financial inclusion, these community members should be provided with knowledge, skills, and tools to improve their existing livelihoods and for managing innovative livelihood collectives and micro-enterprises. Providing access to financial services to society's most vulnerable group in a cost-effective manner through mainstream financial institutions and allowing the poor to become preferred clients of the banking system is fundamental to the financial inclusion strategy of a poverty alleviation programme (Fig 2: C5). The SHG-bank linkage enables an easy access to micro-finance for the SHGs. It also serves to foster their faith towards the banking system.

Good governance systems and processes are crucial to building sensitive support structures and strong institutions of the poor (Fig 2: C6). A well-structured process for participatory identification of the poor by the community helps identify very poor, poor, vulnerable, tribal, differently-abled, and other marginalised communities in a village. A robust process for grading the quality of SHGs and their federations could help maintain a high standard for these institutions. Strong, robust, and transparent monitoring mechanisms (Fig 2: C7) could ensure good governance systems and processes. The process-oriented approach of the programme needs to undergo continuous review, assessment, and course-correction from the qualitative and quantitative progress achieved at various levels. Hence, participants suggested that a robust ICT-based monitoring and evaluation system remain in place for facilitating informed decision-making at all levels. The participants also indicated the urgency of robust implementation of institutional accountability and a self-monitoring process in institutions of the poor at all levels, including peer internal review mechanisms, external social auditing, public expenditure tracking, and community scorecards, in order to build stronger institutions of the poor (Fig 2: C8). Transparency in the functioning of human resources at all levels aided by regular meetings, reviews, and monitoring of progress could ensure effective implementation of the programme. Maintaining equity and transparency in releasing finances and ensuring effective fund utilisation across all eligible groups could also help focus on the most vulnerable groups.

The participants believed that a poverty alleviation programme should have a strong convergence with other welfare programmes (Fig 2: C9). Stronger emphasis should be placed on convergence for developing synergies directly and through the institutions of the poor. Participants suggested that the programme recognise the importance of engaging with industries to set up platforms for public-private-partnerships in farm and non-farm sectors while developing various sector-specific value chains to harness the comparative advantage of the micro-enterprise sector. The political will to support and encourage CBOs, enabling policies for smooth and efficient working of the institutions of the poor, diminished political influence in the decision-making of CBOs, and timely and adequate resource allocation on the part of government institutions is critical for poverty alleviation programmes (Fig 2: C10).

Participants acknowledged that livelihood augmentation requires customised need-based financing for the poor and vulnerable (Fig 2: C11). Access to micro-finance at affordable rates of interest coupled with desired amounts and convenient repayment terms are needed for the poverty reduction of communities. Providing interest subvention for all SHG loans availed from mainstream financial institutions, based on prompt loan repayment, helps develop a healthy loan repayment culture (Fig 2: C12).

Participants opined that diversification of livelihoods would ensure steady incomes for households (Fig 2: C13). The development of micro-enterprise in farm and non-farm sectors could encourage institutions of the poor in the aggregation of produce, value-addition, and marketing of finished goods. Therefore, it is imperative that more and more sustainable enterprises be created by the poor to improve their livelihood security. The demand-driven entrepreneurship (Fig 2: C14) programmes could be taken up through public-private-partnerships. Provisions could be made for incubation funds and start-up funds for the development of multi-sectoral livelihood collectives (Fig 2: C15) to foster a collective entrepreneurship spirit. Livelihood activities, in order to be commercially viable, would require economy of scale, enabling the adoption of available technologies while providing better bargaining power, offering a more significant political clout, and influencing public policy over time. Building specialised multi-sectoral collective institutions of the poor, such as producers' companies and co-operatives could make the latter key players in the market. These livelihood institutions could carry out participatory livelihood mapping and integrated livelihood planning as well as build robust livelihood clusters, supply chains, and value chains. They could also identify gaps in the supply and value chains, create backward and forward linkages, and tap market opportunities for intervention and collectivisation for chosen livelihood activities (Fig 2: C16; C17). Developing adequate and productive infrastructure for processing, storage, packaging, and transportation is crucial for value addition (Fig 2: C16). The demand-based value chain development is currently evident in micro-investment planning processes. Identifying non-farm activities to support enterprises in a comprehensive way could also be crucial. Adequate market linkages and support services like branding, market research, market knowledge, market infrastructure, and backward linkages would go a long way in deriving optimum returns from the chosen livelihood activities (Fig 2: C17).

Several eco-friendly, climate-smart, and innovative approaches in agriculture production systems will ensure the sustainability of production systems even in the context of climate change (Fig 2: C18). Contemporary grassroots innovations supplemented by robust scientific analysis, mainly supported by various government programmes, are likely to ensure enhanced and efficient production systems. Focus on developing adequate infrastructure for processing, storing, and transporting for value addition would serve to reduce post-harvest losses.

Participants believed that affordable and approachable quality education up to the secondary level as well as affordable and quality healthcare facilities are crucial for poverty alleviation (Fig 2: C19). Convergence with mid-day-meal schemes will not only encourage communities

to send their children to schools but also help curb malnutrition. An affordable and approachable healthcare system is likely to help reduce health-related vulnerabilities of the poor. Crucial is an approach that identifies all needy and poor households while primarily focussing on vulnerable sections like scheduled castes, scheduled tribes, particularly vulnerable tribal groups, single women and women-headed households, disabled, landless, migrant labor, isolated communities, and those living in disturbed areas. Equally crucial is including them in institutions of the poor (Fig 2: C20). Customised micro-financing coupled with adequate instruments on healthcare and education could aid vulnerability reduction (Fig 2: C23). The social, human, and personal assets created by developing institutions of the poor are crucial for sustaining and scaling-up of the poverty alleviation programme (Fig 2: C21). This will also allow women to articulate their problems and improve their self-confidence, enhance their respect in society, develop leadership qualities, inspire them to speak and express their feelings unhesitatingly, and empower them economically and socially. Developing an academic understanding of the factors that support community institutions is crucial for the social infrastructure developed to facilitate the social capital building of the poor and vulnerable communities (Fig 2: C22).

### 3.3. FCM-based simulations

In order to evaluate critical factors responsible for poverty alleviation, we used six input vectors for FCM-based simulations. For each scenario, causal propagation occurs in each iteration until the FCM system converges [62–65,67,70,91]. This happens when no change takes place in the values of a concept after a certain point, also known as the system steady-state; the conceptual vector at that point is called the final state vector [62–65,67,70,91]. Values of the final state vectors depend on the structure of the FCM system and concepts considered for input vectors. The larger the value of the final state vectors, the better the selected policies [62–65]. Comparisons between the final state vectors of the alternative simulations are drawn in order to assess the extent of the desired transition by activating each set of input vectors. The initial values and final state vectors of all the concepts for every scenario are presented in Table 2. The graphical representation of various scenarios for poverty alleviation is provided in the S5 Fig.

The first scenario highlights the effects of high-quality community organisations based micro-financing approach. If strong institutions of the poor are built and community heroes start driving the poverty alleviation programme, capacity building of the CBOs gets underway. If mainstream financial institutions start supporting CBOs while customised need-based finance and a repayment culture is developed significant efforts would still be required for putting good governance systems and processes in place along with linkages/ convergences/ partnerships along with other schemes while building capabilities of the poor. In the case of successful micro-financing, there will be opportunities for livelihood diversification, entrepreneurship, multi-sectoral collective enterprise development, value addition by collectives, and market linkages.

The second scenario highlights the effects of the capabilities approach and social security. In this case, affordable and approachable education and healthcare, social inclusion, the building of personal assets, adequate knowledge base, and vulnerability reduction are ensured. In this context, ample efforts will be required for mainstream financial institutions supporting CBOs, good governance systems and processes, and linkages/ convergences/ partnerships with other schemes. Efforts will also be required for a quality support structure and customised need-based finance. The capability and social security enhancement could have prospects for strong institutions of the poor, better implementation processes, livelihood diversification, entrepreneurship, value addition by collectives, multi-sectoral collective enterprise development, and vulnerability reduction.

**Table 2. Initial and final values of all the concepts for every FCM-based scenarios.**

| Concepts | Concept type* | Scenario 1 | | Scenario 2 | | Scenario 3 | | Scenario 4 | | Scenario 5 | | Scenario 6 | |
|---|---|---|---|---|---|---|---|---|---|---|---|---|---|
| | | Initial value | Final value | Initial value | Final value | Initial value | Final value | Initial value | Final value | Initial value | Final value | Initial value | Final value |
| C1: Sensitive support structure | O | 0 | 0.775 | 0 | 0.775 | 0 | 0.775 | 0 | 0.800 | 0 | 0.775 | 0 | 0.775 |
| C2: Strong institutions of the poor | O | 1 | 0.998 | 0 | 0.995 | 0 | 0.989 | 0 | 0.992 | 1 | 0.996 | 1 | 0.996 |
| C3: Communities heroes driving the programme | T | 1 | 1 | 0 | 0 | 0 | 0 | 0 | 0 | 1 | 1 | 1 | 1 |
| C4: Continuous capacity building of the CBOs | O | 1 | 0.866 | 0 | 0.866 | 0 | 0.866 | 0 | 0.896 | 0 | 0.880 | 0 | 0.880 |
| C5: Mainstream financial institutions supporting CBOs | O | 1 | 0.837 | 0 | 0.659 | 0 | 0.659 | 0 | 0.659 | 0 | 0.659 | 0 | 0.659 |
| C6: Good governance systems and processes | O | 0 | 0.659 | 0 | 0.659 | 0 | 0.659 | 1 | 0.837 | 1 | 0.659 | 1 | 0.659 |
| C7: Strong monitoring mechanism | T | 0 | 0 | 0 | 0 | 0 | 0 | 1 | 1 | 0 | 0 | 0 | 0 |
| C8: Implementation process | O | 0 | 0.900 | 0 | 0.950 | 0 | 0.899 | 1 | 0.912 | 0 | 0.900 | 0 | 0.900 |
| C9: Linkages/ Convergences/ Partnerships | O | 0 | 0.659 | 0 | 0.659 | 0 | 0.659 | 1 | 0.821 | 1 | 0.821 | 1 | 0.821 |
| C10: Enabling policy & political will | T | 0 | 0 | 0 | 0 | 0 | 0 | 1 | 1 | 1 | 1 | 1 | 1 |
| CC: People coming out of poverty | R | 0 | 0.999 | 0 | 1 | 0 | 0.999 | 0 | 0.999 | 0 | 1 | 0 | 1 |
| C11: Customized need-based finance | O | 1 | 0.823 | 0 | 0.795 | 0 | 0.795 | 0 | 0.795 | 0 | 0.795 | 0 | 0.795 |
| C12: Developing repayment culture | T | 1 | 1 | 0 | 0 | 0 | 0 | 0 | 0 | 0 | 0 | 0 | 0 |
| C13: Livelihood diversification | O | 0 | 0.987 | 0 | 0.987 | 1 | 0.994 | 0 | 0.987 | 0 | 0.987 | 1 | 0.987 |
| C14: Entrepreneurship | O | 0 | 0.908 | 0 | 0.952 | 1 | 0.905 | 0 | 0.908 | 1 | 0.953 | 0 | 0.953 |
| C15: Multi-sectoral collective enterprise development | O | 0 | 0.996 | 0 | 0.996 | 1 | 0.996 | 0 | 0.997 | 0 | 0.997 | 0 | 0.997 |
| C16: Value addition by collectives | O | 0 | 0.913 | 0 | 0.913 | 1 | 0.913 | 0 | 0.913 | 0 | 0.913 | 0 | 0.913 |
| C17: Market linkages | T | 0 | 0 | 0 | 0 | 1 | 1 | 0 | 0 | 0 | 0 | 0 | 0 |
| C18: Climate smart production systems | O | 0 | 0.816 | 0 | 0.816 | 0 | 0.816 | 0 | 0.816 | 0 | 0.816 | 0 | 0.816 |
| C19: Affordable and approachable education and healthcare | T | 0 | 0 | 1 | 1 | 0 | 0 | 0 | 0 | 1 | 1 | 1 | 1 |
| C20: Social inclusion | T | 0 | 0 | 1 | 1 | 0 | 0 | 0 | 0 | 0 | 0 | 0 | 0 |
| C21: Building of personal assets | O | 0 | 0.659 | 1 | 0.826 | 0 | 0.659 | 0 | 0.659 | 0 | 0.826 | 0 | 0.826 |
| C22: Adequate knowledge base | T | 0 | 0 | 1 | 1 | 0 | 0 | 0 | 0 | 0 | 0 | 0 | 0 |
| C23: Vulnerability reduction | O | 0 | 0.797 | 1 | 0.903 | 0 | 0.797 | 0 | 0.797 | 0 | 0.903 | 0 | 0.903 |

*O = Ordinary; T = Transmitter; R = Receiver

The third scenario highlights the outcomes of the market-based approach. Here, livelihood diversification, entrepreneurship, multi-sectoral collective enterprise development, value addition by collectives, and market linkages are activated. In such a situation, adequate efforts will be required for mainstream financial institutions supporting CBOs, good governance systems

and processes, and linkages/ convergences/ partnerships with other schemes. Efforts will also be required for continuous capacity building of the CBOs, customised need-based finance, affordable and approachable education and healthcare, and vulnerability reduction.

The fourth scenario highlights the outcomes of good governance. Here, good governance systems and processes, robust monitoring mechanisms, implementation processes, enabling policies and political will, and linkages/ convergence/ partnership with other governmental schemes are ensured. In such a situation, plentiful efforts will be required for mainstream financial institutions to lend their support to CBOs and for the building of personal assets. Efforts will also be required for developing a repayment culture, climate-smart production systems, and vulnerability reduction. Good governance is likely to ensure strong institutions of the poor, development of collective enterprises, livelihood diversification, entrepreneurship, value addition by collectives, and market linkages.

In the fifth and sixth scenarios, we activated the most important concepts identified by the participants. The sixth scenario is similar to the fifth one except that the concept C14: entrepreneurship has been replaced by the concept C13: livelihood diversification. The simulation results reveal that quality of CBOs, strong institutions of the poor, community heroes driving the programme, good governance systems and processes, convergence with other schemes/ programmes, enabling policies and political will, and livelihood diversification are very critical for poverty alleviation in a developing nation.

The participants judged a relatively higher weight for the concept C1 (sensitive support structure) (Fig 1). This could be attributed to a conflict of interest on the part of the participants. Even after activating the concept C1 across policy scenarios 4 to 6, the outcome does not change. This also justifies the fact that any community-demand-driven and community-managed poverty alleviation programme has to be self-sustainable in the long-term. Therefore, while a poverty alleviation programme may make use of a support structure in its initial phase, it should persist at thriving even after the support structure has been withdrawn.

## 4. Contributions to FCM and poverty literature and future research directions

This section deals with contributions of the paper to FCM and poverty literature while offering a practical approach to address multi-dimensional poverty. The paper makes a two-fold contribution to FCM literature: i) knowledge capture and sample adequacy and ii) robustness of the dynamic system model. FCM sampling is often extended if additional maps keep adding new dimensions/ insights. The saturation of FCM sampling is formally measured by tracking the number of new concepts introduced in subsequent exercises and estimating an accumulation curve of concepts. When the point of saturation is reached data collection is stopped. In most studies, the saturation of FCM sampling is reported at 30–32 maps [60,62,63,66,72]. This study does demonstrate, however, that in the event of a 'mixed-concept design' approach when the participants gain access to concepts already identified by other sets of participant groups the latter participants continue to add new concepts, making the system much more complex and the data richer. Most FCM-based case studies published in scientific journals have taken weights of the causal interactions between the concepts. This study has not only obtained weights of the causal interactions between the concepts but also obtained weights of each concept. Results of the FCM-based simulations, by and large, match with the most critical concepts identified by participants represented by higher relative weights. This demonstrates in-depth understanding of participants of the subject matter and robustness of the system.

Scenarios are defined as 'a plausible description of how the future may develop based on a coherent and internally consistent set of assumptions' [97]. It also represents uncertainty as a range of plausible futures. Hence, in order to establish proper causal pathways of various poverty eradication approaches, it may be necessary to design random control trial experiments along each of the poverty eradication approaches and carry out the efficacy of each approach delineated above using the *difference-in-difference* micro-econometric model.

## 5. Conclusions

The results of our FCM-based simulations reveal that in order to eradicate poverty one needs to provide micro-finance through high-quality community organisations, enhance capabilities of the poor while providing social safety nets to the poor and vulnerable, ensure good governance within community organisations and institutions supporting them, continue to diversify livelihood options, and provide market linkages to small producers. Our findings confirm that various approaches to poverty alleviation are rather complementary and need to be implemented simultaneously for a comprehensive poverty alleviation drive. However, in relative terms, factors like good governance within community organisations and supporting institutions, high-quality community organisations based micro-financing, and enhancement of capabilities coupled with social security assurance seem to work better than a market-based approach. There is rich literature available on radical approaches like land reforms, decentralisation and poverty alleviation that have not been evaluated in this study. Nevertheless, findings of the study lead us to conclude that in order to address multi-dimensional poverty an integrated and multi-dimensional poverty alleviation approach is needed. Findings of the study are likely to help improve the design, management, and implementation of poverty eradication programmes in developing countries.

## Supporting information

**S1 Fig. A simple FCM obtained during the workshop conducted in phase 1.**
(PDF)

**S2 Fig. The survey instrument used during phase 2 of the data collection.**
(PDF)

**S3 Fig. A list of all the concept categories and sub-concepts after data collection.**
(PDF)

**S4 Fig. Sensitivity analysis of the system.**
(PDF)

**S5 Fig. Various scenarios for poverty alleviation.**
(PDF)

**S1 Table. Social cognitive map.**
(XLSX)

## Acknowledgments

We thank the World Bank team and functionaries of DAY-NRLM at national, state, and district levels for participating in the study. Indrani Talukdar is acknowledged for language editing. We thank the Academic Editor and the two anonymous reviewers for providing insightful comments and constructive suggestions.

## Author Contributions

**Conceptualization:** Pramod K. Singh.

**Data curation:** Harpalsinh Chudasama.

**Formal analysis:** Pramod K. Singh.

**Funding acquisition:** Pramod K. Singh.

**Methodology:** Pramod K. Singh.

**Writing – original draft:** Pramod K. Singh.

**Writing – review & editing:** Pramod K. Singh.

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
