## [Decision Letter · Decision Letter 0]

14 Nov 2019

PONE-D-19-26538

Evaluating Poverty Alleviation Strategies in a Developing Country

PLOS ONE

Dear Dr. Singh,

Thank you for submitting your manuscript to PLOS ONE. After careful consideration, we feel that it has merit but does not fully meet PLOS ONE’s publication criteria as it currently stands. Therefore, we invite you to submit a revised version of the manuscript that addresses the points raised during the review process.

The manuscript requires further revisions are regards the abstract, the contribution to the related literature, literature, research methodology, outcomes’ reporting style. Not least, an extensive English language is required.

We would appreciate receiving your revised manuscript by Dec 22 2019 11:59PM. To enhance the reproducibility of your results, we recommend that if applicable you deposit your laboratory protocols in protocols.io, where a protocol can be assigned its own identifier (DOI) such that it can be cited independently in the future. For instructions see: http://journals.plos.org/plosone/s/submission-guidelines#loc-laboratory-protocols

We look forward to receiving your revised manuscript.

Kind regards,

Stefan Cristian Gherghina, PhD

Academic Editor

PLOS ONE

Journal Requirements:

1.

Thank you for including your ethics statement:  "Research Ethics Committee of our Institute approved the methodology of this paper, including the neutral problem domain— direct and consequential impacts of deforestation.

Collection of plant, animal, or other materials are not involved in this study"

Reviewers' comments:

Reviewer's Responses to Questions

**Comments to the Author**

1. Is the manuscript technically sound, and do the data support the conclusions?

Reviewer #1: Yes

Reviewer #2: Yes

2. Has the statistical analysis been performed appropriately and rigorously? 

Reviewer #1: Yes

Reviewer #2: Yes

3. Have the authors made all data underlying the findings in their manuscript fully available?

Reviewer #1: Yes

Reviewer #2: Yes

4. Is the manuscript presented in an intelligible fashion and written in standard English?

Reviewer #1: No

Reviewer #2: Yes

5. Review Comments to the Author

Reviewer #1: The is an interesting research which identifies the critical factors for poverty alleviation in India with the aid of fuzzy cognitive maps (FCMs). This paper has many strengths and some opportunities for improvement, which I will elaborate below:

Abstract has inappropriate structure. I suggest to answer the following aspects: - general context - novelty of the work - methodology used - main results

Section 1 presents interesting information. However, it fails to set out any specific interest to a broader audience. There is nothing more than a sort of putting forward the topic. However, what about contribution to relevant literature? Which gaps do you want to fill and how?

Methodology is unclear. Initially a short resume can be proposed to explain several steps. The methodology used must be linked to the existing literature on FCM. what is its potential? its limit?

Results must be linked to the methodology. Please define the relationship and relate your finding with the relevant literature.

Finally, an extensive editing of English language and style is required.

Suggested references:

https://doi.org/10.1016/j.techfore.2019.07.012

https://doi.org/10.1016/j.eist.2015.06.006

https://doi.org/10.1016/j.jenvman.2015.10.038

Reviewer #2: The paper is accurate in the description of the methodology; however some steps can be explained better.

In 189-190 you explained that the concepts were elicited asking the participants the “critical 190 factors required to ensure that people come out of poverty on a sustainable basis”. Was this enough to prompt the contribution of the participants or did you give some other information to elicit their contribution. The quantity of information given before the participant tasks is a question that matter in FCMs building since a great quantity of information could lead to bias while very little information can lead to scanty results. How did you reach the correct trade-off?

523-524 “The 524 larger the value of the final state vectors, better the selected policies.” This means that all the concept give a desired and positive contribution to the poverty alleviation. Did all respondent give positive concept or did you declined all in a positive way to make them handy?

The paper aims also at giving a methodological contribution. I suggest some recent paper to enrich this part:

Falcone, P. M., Lopolito, A., & Sica, E. (2019). Instrument mix for energy transition: A method for policy formulation. Technological Forecasting and Social Change, 148, 119706.

Morone, P., Falcone, P. M., & Lopolito, A. (2019). How to promote a new and sustainable food consumption model: A fuzzy cognitive map study. Journal of cleaner production, 208, 563-574.

Falcone, P. M., Lopolito, A., & Sica, E. (2018). The networking dynamics of the Italian biofuel industry in time of crisis: Finding an effective instrument mix for fostering a sustainable energy transition. Energy Policy, 112, 334-348.

Falcone, P. M., Lopolito, A., & Sica, E. (2017). Policy mixes towards sustainability transition in the Italian biofuel sector: Dealing with alternative crisis scenarios. Energy research & social science, 33, 105-114.

ln 192 why do you mention only two concepts?

The diagrams of the various scenarios are hard to read. I suggest bar diagrams showing differences with the steady state values of each variable under each scenario.

6. PLOS authors have the option to publish the peer review history of their article (what does this mean?). If published, this will include your full peer review and any attached files.

Reviewer #1: No

Reviewer #2: No

---

## [Author Response · Author response to Decision Letter 0]

3 Dec 2019

Uploaded as separate file namely, Response to Reviewers

---

## [Decision Letter · Decision Letter 1]

16 Dec 2019

Evaluating Poverty Alleviation Strategies in a Developing Country

PONE-D-19-26538R1

Dear Dr. Singh,

We are pleased to inform you that your manuscript has been judged scientifically suitable for publication and will be formally accepted for publication once it complies with all outstanding technical requirements.

With kind regards,

Stefan Cristian Gherghina, PhD

Academic Editor

PLOS ONE

Additional Editor Comments (optional):

Reviewers' comments:

Reviewer's Responses to Questions

**Comments to the Author**

1. If the authors have adequately addressed your comments raised in a previous round of review and you feel that this manuscript is now acceptable for publication, you may indicate that here to bypass the “Comments to the Author” section, enter your conflict of interest statement in the “Confidential to Editor” section, and submit your "Accept" recommendation.

Reviewer #1: All comments have been addressed

Reviewer #2: All comments have been addressed

2. Is the manuscript technically sound, and do the data support the conclusions?

Reviewer #1: Yes

Reviewer #2: Yes

3. Has the statistical analysis been performed appropriately and rigorously? 

Reviewer #1: Yes

Reviewer #2: N/A

4. Have the authors made all data underlying the findings in their manuscript fully available?

Reviewer #1: Yes

Reviewer #2: Yes

5. Is the manuscript presented in an intelligible fashion and written in standard English?

Reviewer #1: Yes

Reviewer #2: Yes

6. Review Comments to the Author

Reviewer #1: The authors accurately addressed all my previous comments. I recommend the article for publication in PLOS ONE Journal.

Reviewer #2: The author(s) has/have addressed all the point raised in the review. The paper has been much improved and suits for the publication in the Journal

7. PLOS authors have the option to publish the peer review history of their article (what does this mean?). If published, this will include your full peer review and any attached files.

Reviewer #1: No

Reviewer #2: No

---

## [Editor Report · Acceptance letter]

20 Dec 2019

PONE-D-19-26538R1 

Evaluating Poverty Alleviation Strategies in a Developing Country 

Dear Dr. Singh:

I am pleased to inform you that your manuscript has been deemed suitable for publication in PLOS ONE. Congratulations! Your manuscript is now with our production department. 

With kind regards,

on behalf of

Dr. Stefan Cristian Gherghina 

Academic Editor

PLOS ONE